# Evaluation of the Saline–Alkaline Tolerance of Rice (*Oryza sativa* L.) Mutants Induced by Heavy-Ion Beam Mutagenesis

**DOI:** 10.3390/biology11010126

**Published:** 2022-01-13

**Authors:** Xin Zhang, Fu Yang, Hongyuan Ma, Jingpeng Li

**Affiliations:** 1Northeast Institute of Geography and Agroecology, Chinese Academy of Sciences, Changchun 130102, China; zhangxin183@mails.ucas.ac.cn (X.Z.); yangfu@iga.ac.cn (F.Y.); mahongyuan@iga.ac.cn (H.M.); 2University of Chinese Academy of Sciences, Beijing 100049, China

**Keywords:** rice, heavy-ion beam, mutant, saline–alkaline stress

## Abstract

**Simple Summary:**

Soil salinization is one of the important obstacles restricting agricultural development. Cultivating new varieties of saline–alkaline-tolerant rice can increase the yield of rice. It is possible to minimize breeding costs and shorten the breeding period through heavy ion beam irradiation mutation breeding. Our research evaluated and screened saline–alkaline-tolerant rice mutants induced by heavy ion beams. The results show that heavy ion beam radiation is an effective method for breeding new saline–alkaline-tolerant rice cultivars, and the selected mutant lines have excellent production performance under saline–alkaline stress. Our research results provide new theoretical and practical insights that can be used to help develop new saline–alkaline-tolerant rice cultivars.

**Abstract:**

Soil salinity is a widespread and important abiotic factor impeding rice production by adversely affecting seed germination, seedling growth, and plant productivity. In this study, the rice cultivar TH899 was treated with 200 Gy of heavy-ion beam irradiation, and 89 mutant lines with stable phenotypes were selected using the pedigree method based on continuous assessment over six years. The seed germination performance of these mutants was tested under different saline–alkaline concentrations. Five highly tolerant lines were further evaluated in a series of experiments at the seedling stage and in the field. During the seedling stage, the reduction of seedling length, root length, fresh weight, and dry weight were dramatically lower in these five mutants than those in TH899 under saline–alkali stress. The K^+^/Na^+^ ratio was higher in these five mutants than in TH899. In the field experiment, the grain yield of mutant lines was higher than that of TH899. In addition, the grain yield of mutant line M89 was higher than that of the local cultivar in actual production. These mutant lines are expected to increase grain yield in soda saline–alkaline regions in northeast China.

## 1. Introduction

Saline–alkaline stress is an important form of abiotic stress impeding agricultural development [1]. Salinization affects more than 1 billion hectares of soil worldwide and induces declines in agricultural productivity of more than 20% of the world’s arable land [2,3]. There are approximately 100 million hectares of salinized land in China, and most of this land is distributed in Northeast, North, and Northwest China as well as coastal areas [4,5]. Saline–alkaline soil in Northeast China is one of the three main areas with saline–alkaline soil worldwide. Soil salinity in this area is mainly caused by Na_2_CO_3_ and NaHCO_3_. Hydrolysis of Na_2_CO_3_ and NaHCO_3_ occurs in the soil, resulting in abnormally high pH values ranging from 8.5 to 11. [2,6]. High salinity and alkalinity can significantly affect crop growth and yields [7].

The world population is expected to reach 9 billion by 2050, and food production will need to increase by at least 70% to meet the projected increase in demand [8,9]. Rice is an important crop that provides food for half of the world’s population [3]; it thus plays an important role in ensuring food security [8]. However, rice is a non-halophyte and thus vulnerable to saline–alkaline stress [10]. Therefore, the breeding of saline–alkaline-tolerant rice cultivars that can grow on saline–alkaline soils is important for increasing food production and ensuring food security.

The breeding of saline–alkaline-tolerant rice has long been a major challenge [11,12,13]. Traditional breeding methods have produced many saline–alkaline-tolerant cultivars through crosses, backcrosses, and repeated selection. However, the traditional breeding of saline–alkaline-tolerant rice cultivars is time-consuming (taking at least 6 to 7 years), laborious, and inefficient [14,15]. Genetic engineering is a powerful tool for breeding saline–alkaline-tolerant rice cultivars [16], which involves directly introducing saline–alkaline-tolerant genes into rice cultivars. Although genetic engineering technology greatly shortens the time required for breeding, public acceptance of transgenic technology is low [17]. Mutagenesis is a convenient technology for breeding saline–alkaline-tolerant rice cultivars, which is easily accepted by the public [18,19]. Mutation breeding is an important tool for creating genetic variability in desired traits [20], including the breeding of new cultivars with higher resistance to biotic and abiotic stress and agronomic traits [21]. Common mutation breeding includes EMS mutagenesis, space mutagenesis, and irradiation mutagenesis [22]. Although the operation of EMS mutagenesis is simple, the mutagenesis frequency is low and the workload is high [23]. Space mutagenesis has high relative biological benefits, but its cost is high and operation is difficult [24]. The radiation mutagenesis method is simple and the mutagenesis frequency is high [25].

Heavy-ion beam radiation is a new crop genetic improvement technology that can effectively alter target traits [26]. Compared with traditional radiation mutagenesis, heavy-ion beam mutagenesis can induce broad-spectrum mutations at high frequencies, exert more pronounced biological effects that result in less damage to target species, and generate mutations that are stably inherited, making it an ideal method for breeding new cultivars [27]. In recent years, an increasing number of researchers have used heavy-ion beam irradiation technology to conduct mutagenesis studies of various plants, including ornamental plants (e.g., chrysanthemum [28] and carnation [29]), economic crops (e.g., tobacco [30] and tomato [31]), and food crops (e.g., sweet sorghum [32] and soybeans [33]). Few studies have used heavy-ion beam irradiation technology to breed new saline–alkaline-tolerant rice cultivars.

The aims of this study were to explore the utility of heavy-ion beam radiation mutagenesis for breeding saline–alkaline-tolerant rice cultivars. Specifically, we sought to address the following questions: (1) Can heavy-ion beam irradiation be used to produce saline–alkaline-tolerant rice mutants? (2) Do the mutant lines produced have higher yields under saline–alkaline stress? Our results provide new theoretical and practical insights that could be used to aid the development of new saline–alkaline-tolerant rice cultivars.

## 2. Materials and Methods

### 2.1. Plant Materials and Irradiation Treatments

The rice cultivar Tonghe899 (hereafter referred to as TH899) was used for irradiation treatment. The average yield of this cultivar was 585.4 kg/mu, and according to the “Edible Rice Variety Quality Inspection Standards” issued by the Ministry of Agriculture, the rice quality reached the second-level requirements, and the cultivar showed resistance to rice blast and leaf spot disease [34]. The irradiation treatments were conducted at the Institute of Modern Physics of the Chinese Academy of Sciences in March 2013. Dried rice seeds of TH899 with uniform size were divided into two groups of 150 seeds each that received different doses of irradiation. Seeds were exposed to irradiation at 20 Gy per min with doses of 0 (control) and 200 Gy. The irradiation was directly targeted at the hilum of each seed with an 80 MeV/u carbon ion beam.

### 2.2. Breeding of the Mutant Population

In April 2013, the irradiated seeds were sown to raise seedlings, and 23 plants survived before transplanting. In autumn, all 23 plants of M1 generation seeds were mixed and harvested. The plants were planted in Hainan in the winter of 2013. A large number of mutant lines appeared in the M2 generation, and 78 mutagenic lines were selected. These mutant lines were bred to the M6 generation using the pedigree method (Figure 1). A total of 89 mutant lines with stable phenotypes for at least two generations were selected for subsequent experiments.

### 2.3. Seed Germination of Mutant Lines under Saline–Alkaline Stress

Based on the properties of saline–alkaline soil in Northeastern China and data from our preliminary experiment, a 90 mmol/L mixed saline–alkaline solution (Na_2_CO_3_:NaHCO_3_ = 1:1) was used for the preliminary screening of mutant lines. First, seeds of the mutant lines were surface sterilized by soaking them in a solution of 2.5% (*w*/*v*) NaClO_3_ for 10 min and then washed thoroughly with sterile water. Fifty seeds were randomly selected and placed evenly in a 9-cm-diameter Petri dish with two layers of filter paper. Ten milliliters of the mixed saline–alkaline solution was added to the treatment group, and 10 mL of sterile water was added to the control group; seeds were then placed in a 30 °C incubator for germination. Each treatment was repeated three times. The number of seeds germinated on the 7th day was determined, and the final germination percentage was calculated [35].

### 2.4. Seedling Growth of Mutant Lines under Saline–Alkaline Stress

Seeds of mutant lines were surface sterilized following the procedures described in the above experiment. The seeds were then placed on filter paper in a Petri dish, moistened with sterile water, and placed in a thermostat at 30 °C for 48 h. Next, 24 uniformly germinated seeds were selected and transferred to PCR plates floating in a tank (48 cm × 36 cm × 10 cm) containing 10 L of deionized water. After 7 d of culture, the Kimura nutrient solution was added instead of deionized water, and the culture was continued for another 7 d. After 14 d of pre-culture, the control group continued to be cultured in Kimura nutrient solution, and the saline–alkaline treatment group was moved to culture medium containing 20 mmol/L mixed saline–alkaline solution (Na_2_CO_3_:NaHCO_3_ = 1:1, screening in pre-test). Each treatment was repeated four times. During culture, the solution was changed every 3 d, the temperature was 26 °C, and the photoperiod was 12 h light/12 h dark.

After 7 d of saline–alkaline stress, a ruler was to measure the height and root length of six rice seedlings. After separating the roots, stems, and leaves of the six seedlings, the fresh weight of the seedlings and roots was measured using a scale accurate to 0.001 g. The seedlings were then placed in parchment paper bags in an oven. Seedlings were dried at 80 °C to constant weight, followed by 105 °C for 30 min to deactivate enzymes. The dry weights of the seedlings and roots were then measured again.

The K^+^ and Na^+^ content was measured following the method of Wang and Zhao [36] with slight modifications. The dried plant samples were ground into a powder with a mortar and passed through a 30-mesh sieve. Next, 0.05-g samples were placed into a test tube with 10 mL of deionized water, and the contents were mixed by shaking. The test tube was then boiled in a 100 °C water bath for 2 h, cooled, and filtered. The K^+^ and Na^+^ content was determined by 6400A flame photometry.

### 2.5. Field Growth of Mutant Lines under Saline–Alkaline Stress

In 2019 and 2020, the experiment was conducted at Da’an Sodic Land Experimental Station of the Chinese Academy of Sciences, which is located in Jilin Province, China (45°35′58″–45°36′28″ N, 123°50′27″–123°51′31″ E, 132.1 m above sea level). The experiment was conducted on two different pieces of land: Field 1 was the control, and Field 2 was the treatment. The physical and chemical properties of the soil are shown in Table 1. The experiment was conducted in a randomized complete block design with four replications. Seedlings were transplanted at a plant spacing of 20 cm and a row spacing of 30 cm, with 3 rows per mutant line and 10 plants per row. Seedlings were raised on April 12 and transplanted on May 29. Water and fertilizer management in the field followed procedures used by local farmers.

The measurement of yield and yield components followed the methods of Yoshida [37]. At the mature stage, the average number of tillers per hill in the quadrat was counted. Plant height (PH), panicle length (PL), panicle number per plant (PN), spikelet per panicle (SP), 1000-grain weight (1000-GW), and percentage of filled spikelet (PFS) were determined by selecting lines with the same number of tillers. Finally, the grain yield of a 1-m^2^ quadrat of each mutant line was measured.

### 2.6. Field Performance of M89

In 2020, the mutant line M89 was certified as a new rice variety, and it became the main recommended variety in 2021. M89 has been promoted to more than 200,000 mu in the saline–alkaline area of Jilin Province, China. In 2021, the experiment was conducted at Da ‘an city, which is located in Jilin Province, China (45°57′00″–45°45′51″ N, 123°08′45″–124°21′56″ E, 132.1 m above sea level). Three field plots for planting M89 and local varieties were randomly selected, and the grain yield was measured by harvesting all plants of each field plot (1000 m^2^) and converting on a hectare basis. All field plots are planted by local farmers

### 2.7. Statistical Analysis

Germination percentage data of 90 rice genotypes with three repeats for each genotype were normalized by arcsine transformation prior and analyzed by two-way analysis of variance (ANOVA). Hierarchical cluster analysis of 90 rice genotypes with three repeats for each genotype based on the Euclidean distance and Ward’s minimum variance method was performed using Origin 2021b. Data of 6 rice genotypes with four repeats for each genotype from the seedling and field experiments were analyzed by one-way ANOVA, and Tukey’s test was used to compare the differences between the rice genotypes (α = 0.05). All data analyses were conducted in R 3.6.3; graphs were constructed using Origin 2021b.

## 3. Results

### 3.1. Seed Germination of Mutant Lines under Saline–Alkaline Stress

Saline–alkaline stress (F_1, 360_ = 1751.68, *p* < 0.01), genotype (F_89, 360_ = 14.65, *p* < 0.01), and the the saline–alkaline stress × genotype interaction (F_89, 360_ = 13.98, *p* < 0.01) had a significant effect on seed germination. Under control conditions, the average germination percentage of the mutant lines was 99.5% (Table 2), which did not significantly differ from that of the parent TH899. Under saline–alkaline stress, the germination percentage of the mutant lines was between 9.3% and 88.7%, and the average germination percentage was 49.5%, which was significantly higher than that of TH899. The germination percentage of 69 mutant lines was higher than that of TH899.

Ward’s method was used for systematic cluster analysis. A total of 89 mutant lines and TH899 were divided into five clusters: Highly tolerant (10), tolerant (20), moderately tolerant (24), sensitive (30), and highly sensitive (6) (Figure 2).

Based on the clustering results, five mutant lines with high tolerance to saline–alkaline stress (M5, M41, M60, M88, and M89) were selected to explore the response of mutant lines to saline–alkaline stress at the seedling stage and in the field.

### 3.2. Seedling Growth of Mutant Lines under Saline–Alkaline Stress

Saline–alkaline stress had an inhibitory effect on the growth of rice seedlings; the seedling length and seedling fresh weight of the mutant lines and TH899 were reduced compared with control conditions (*p* < 0.05). Seedling length and seedling fresh weight of the mutant lines were decreased by 23.13% to 33.20% and 27.67% to 47.55%, respectively, under saline–alkaline stress compared with control conditions; seedling length and seedling fresh weight of TH899 were reduced by 37.35% and 55.10%, respectively, under saline–alkaline stress compared with control conditions (Figure 3a,c). Saline–alkaline stress had no significant effect on the root length and root fresh weight of the mutant lines (*p* > 0.05) but it had a significant inhibitory effect on these traits in TH899 (*p* < 0.05, Figure 3b,d). Saline–alkaline stress had no significant effect on the seedling dry weight and root dry weight of the mutant lines and TH899 (*p* > 0.05). The seedling dry weight and root dry weight of the mutant lines were decreased by 3.63% to 12.20% and 0.79% to 6.48%, respectively, under saline–alkaline stress compared with control conditions; the seedling dry weight and root dry weight of TH899 were reduced by 11.44% and 10.68%, respectively, under saline–alkaline stress compared with control conditions (Figure 3e,f).

### 3.3. K^+^ and Na^+^ Content in Mutant Lines under Saline–Alkaline Stress

There were significant differences in the K^+^ and Na^+^ content and K^+^/Na^+^ ratio between mutant lines and TH899 under saline–alkaline stress. Under saline–alkaline stress, the K^+^ content in the roots of the mutant lines ranged from 15.07 mM/g to 21.23 mM/g, which was higher than that of TH899 (12.97 mM/g); the Na^+^ content ranged from 24.23 mM/g to 27.93 mM/g, which was lower than that of TH899 (32.67 mM/g) (Table 3). The K^+^/Na^+^ ratio ranged from 0.58 to 0.76, which was higher than that of TH899 (0.40).

### 3.4. Grain Yield and Yield Components of Mutant Lines under Saline–Alkaline Stress

The grain yield of the mutant lines and TH899 was reduced under saline–alkaline stress compared with the control (Figure 4a,b). In 2019, the grain yield of the mutant lines was decreased by 3.6% to 31.5%, and the grain yield of TH899 was decreased by 11.7%. Under control conditions and saline–alkaline stress, the grain yield of M88 and M89 was higher than TH899(Figure 4a). In 2020, the grain yield of the mutant lines was decreased by 31.2% to 50.2%, and the grain yield of TH899 was significantly decreased by 61.4%. Under control conditions and saline–alkaline stress, the grain yields of M5, M60, and M89 were higher than TH899 (Figure 4b). Overall, the grain yield of M89 is the highest regardless of the control conditions or the saline–alkaline stress in 2019 and 2020, and the average grain yield was increased by 41.2% compared with TH899 under saline–alkaline stress.

In 2019, the ANOVA of the yield components of the mutant lines and TH899 under saline–alkaline stress revealed significant differences in SP and 1000-GW between the mutant lines and TH899. The SP and the 1000-GW were higher in mutant lines compared with TH899 (Table 4). In 2020, ANOVA of the yield components of the mutant lines and TH899 under saline–alkaline stress revealed significant differences in PH, PL, and 1000-GW between the mutant lines and TH899. The PH was shorter (84.8 cm to 95.0 cm), the PL was longer (17.5 cm to 19.8 cm), and the 1000-GW was higher (22.02 g to 25.30 g) in mutant lines compared with TH899 (Table 5).

### 3.5. Field Performance of M89

In a large-scale field experiment, the grain yield of M89 was 9480 kg/ha, which is 10.6% higher than the local main varieties, and the grain yield of high-yield plots reached 10,815 kg/ha. The mutant line M89 showed higher performance in a large-scale field experiment, which indicated that this line could be used as a germplasm resource to improve the saline–alkaline tolerance of rice.

## 4. Discussion

Soil salinization is one of the most important types of abiotic stress impeding rice production, as it can result in 20% to 100% reductions in rice yields [12]. The potential for breeding new saline–alkaline-tolerant rice varieties is high given that there is extensive genetic variation in saline–alkaline stress tolerance in rice [38]. Mutation breeding is a strategy that can be used to cultivate new saline–alkaline-tolerant rice varieties; the first goal of mutation breeding is to generate high–yielding elite varieties with ideal traits that are vulnerable to saline–alkaline stress to create variation in saline–alkaline tolerance [20,39]. Under normal soil conditions, the rice variety used in this study (TH899) produces high yields, possesses excellent quality traits, and shows strong disease resistance.

Mutagenesis is one of the main strategies for obtaining saline–alkaline-tolerant genotypes; it is thus an important tool for improving the tolerance of rice to saline–alkaline stress [20,40]. We successfully screened 63 mutant lines that had a higher seed germination percentage than the parent TH899 under saline–alkaline stress (Table 2, Figure 2). The results indicated that heavy-ion beam irradiation successfully induced variation in saline–alkaline tolerance and was an effective method for developing saline–alkaline-tolerant varieties. Similarly, Song, et al. [41] successfully isolated saline–alkaline-tolerant mutant lines by gamma-ray irradiation.

We also analyzed the growth and development of the seedlings of mutant lines and TH899. Previous studies have shown that saline–alkaline stress can significantly inhibit the growth and development of rice seedlings [42]. The results of this study indicate that the seedling length, root length, fresh weight, and dry weight of seedlings were altered under saline–alkaline stress, but the observed responses of mutant lines and TH899 to saline–alkaline stress differed (Figure 3). This may stem from the fact that the mutant lines and TH899 had different saline–alkaline stress tolerance mechanisms. Each rice variety displays one or two salt-tolerance mechanisms [11], which mitigate the damage induced by saline–alkaline stress by regulating physiological and biochemical mechanisms [17,42,43].

Rice saline–alkaline tolerance is a complex trait involving multiple physiological and biochemical mechanisms [11,44,45]. Cationic transport is one of the main mechanisms regulating saline–alkaline tolerance in rice; it contributes to the maintenance of ion homeostasis under saline–alkaline stress by expelling Na^+^ and absorbing K^+^ [45]. We found that the mutant lines can control the uptake of these ions in the roots by reducing the Na^+^ concentration and absorbing more K^+^, which increases the K^+^/Na^+^ ratio (Table 3). This is consistent with the findings of Nakhoda, et al. [46]. Therefore, the increased saline–alkaline tolerance of the mutant line seedlings might stem from their ability to maintain a higher K^+^/Na^+^ ratio in the root system.

The saline–alkaline tolerance in the early stage of rice is not always related to the saline–alkaline tolerance of rice during the entire growth period [13,47]. For this reason, we characterized the saline–alkaline tolerance of rice during the entire growth period in the field. We found that the grain yield of M89 was higher than that of TH899 under saline–alkaline stress (Figure 4). The results indicated that heavy-ion beam irradiation can produce high-yield and saline–alkaline-tolerant mutant lines of rice. These results were consistent with those of Domingo, et al. [48], wherein saline–alkaline-tolerant lines with higher yield were isolated through radiation mutagenesis. In this study, PH, SL, SP, and 1000-GW significantly differed among mutant lines (Table 4 and Table 5). The results indicated that heavy-ion beam irradiation successfully induced variation in these agronomic traits, most likely by affecting the genes controlling these traits.

## 5. Conclusions

In conclusion, we successfully induced and evaluation some saline–alkaline-tolerant mutant lines by heavy-ion beam irradiation, and the saline–alkaline-tolerant varieties of these lines were better than the parent TH899. Five highly tolerant lines were further evaluated in a series of experiments at the seedling stage and in the field. At seedling stage, the highly tolerant lines absorbed more K^+^ and limited Na^+^ absorption, resulting in a higher K^+^/Na^+^ ratio in the roots. In the field, the highly tolerant lines had higher grain yield than the parent TH899. In addition, the grain yield of the mutant line M89 was higher than that of the local cultivar in actual production.

## Figures and Tables

**Figure 1 biology-11-00126-f001:**
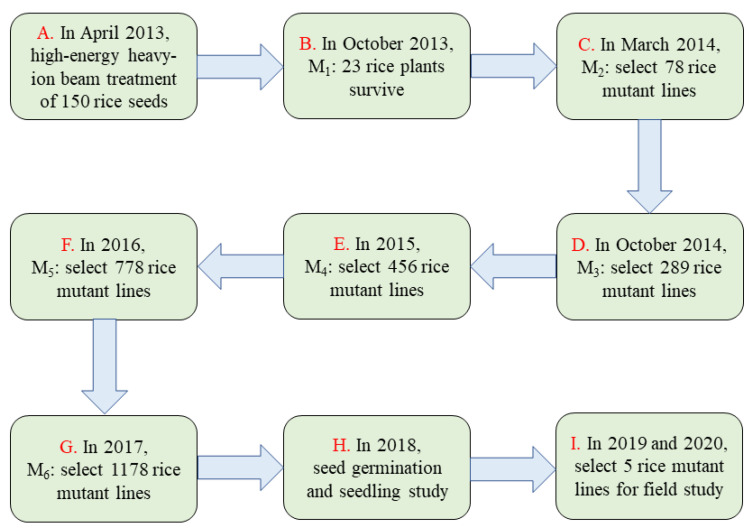
Development of the heavy-ion beam mutagenesis rice population. Schematic diagram of the steps that were taken to create (**A**), breed (**B**–**G**), and test (**H**–**I**) the mutant lines. M1–M6 represent the mutant generation.

**Figure 2 biology-11-00126-f002:**
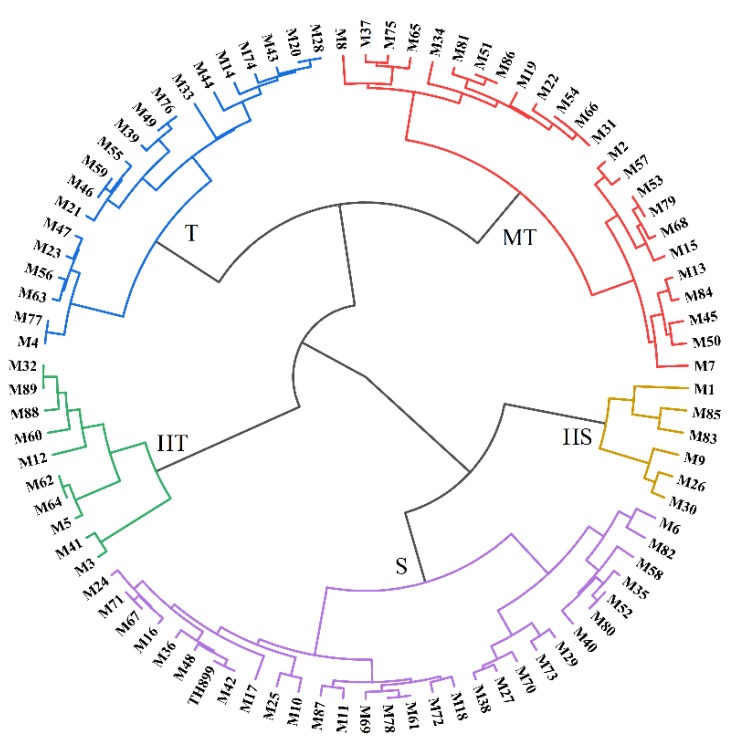
Hierarchical cluster analysis based on the germination percentage. Euclidean distance and Ward’s algorithm were used to evaluate the saline–alkaline tolerance of 89 mutant lines and the parent TH899. Green indicates highly tolerant (HT) lines; blue indicates tolerant (T) lines; red indicates moderately tolerant (MT) lines; purple indicates sensitive (S) lines; and yellow indicates highly sensitive (HS) lines.

**Figure 3 biology-11-00126-f003:**
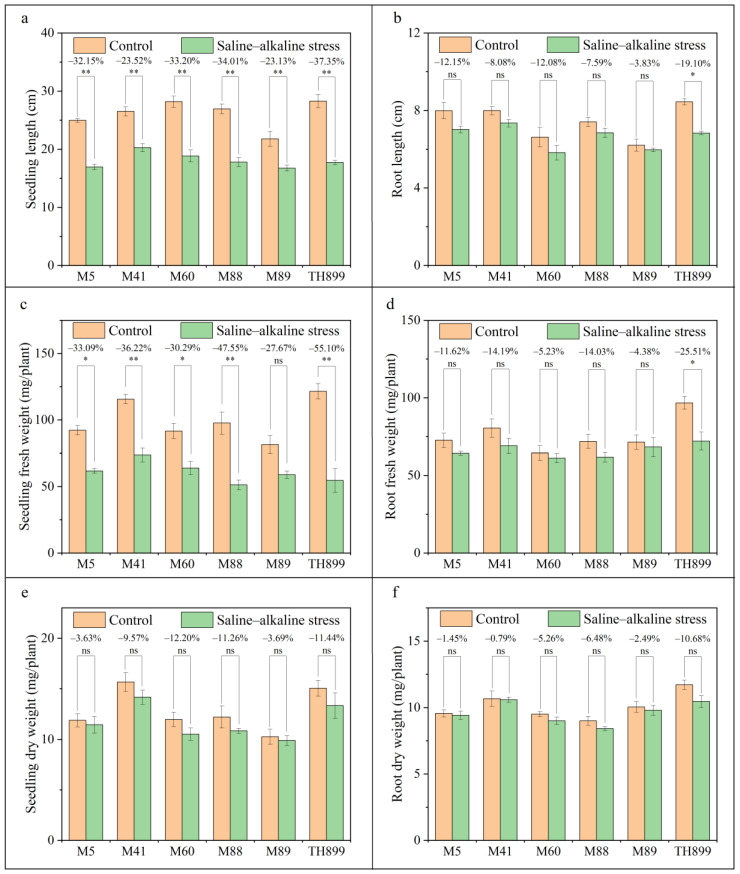
Seedling growth and biomass of mutant lines and the parent TH899 under control and saline–alkaline stress conditions. (**a**) Seedling length, (**b**) root length, (**c**) seedling fresh weight, (**d**) root fresh weight, (**e**) seedling dry weight, and (**f**) root dry weight. Bars are the means and standard errors of four replications. ** indicates significant differences at *p* < 0.01; * indicates significant differences at *p* < 0.05; and ns indicates no significant differences. The values on top of the bars indicate the decrease in each agronomic trait under saline–alkaline stress relative to the control.

**Figure 4 biology-11-00126-f004:**
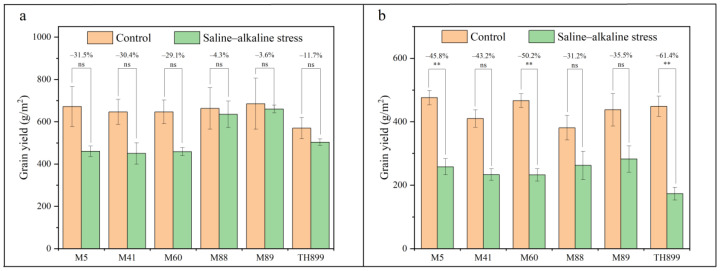
Grain yield of mutant lines and the parent TH899 under saline–alkaline stress and control conditions in 2019 (**a**) and 2020 (**b**). Bars are the means and standard errors of three (**a**) and four (**b**) replications. ** indicates significant differences at *p* < 0.01; and ns indicates no significant differences. The values on top of the bars indicate the decrease in grain yield under saline–alkaline stress relative to the control.

**Table 1 biology-11-00126-t001:** Physical and chemical properties of the soil at the experimental site.

Type	pH	EC (dS/m)	CEC (cmol/kg)	Na^+^ (cmol/kg)	ESP (%)
Field 1	8.3	0.31	37.41	3.85	10.6
Field 2	9.2	0.48	27.66	9.15	32.8

**Table 2 biology-11-00126-t002:** Seed germination of mutant lines and TH899 under control and saline–alkaline stress conditions.

G	GP (%)		G	GP (%)		G	GP (%)	
	Control	Stress		Control	Stress		Control	Stress
M1	96.7	54	M31	100	48.7	M61	100	39.3
M2	100	48.7	M32	100	81.3	M62	100	76
M3	99.3	36	M33	100	66	M63	100	68.7
M4	100	81.3	M34	100	46	M64	100	76.7
M5	99.3	88.7	M35	98.7	27.3	M65	100	44.7
M6	100	19.3	M36	100	37.3	M66	100	48.7
M7	98.7	53.3	M37	97.3	45.3	M67	100	35.3
M8	99.3	61.3	M38	100	30.7	M68	100	55.3
M9	100	15.3	M39	99.3	64	M69	99.3	40
M10	99.3	41.3	M40	100	26	M70	100	28.7
M11	98.7	40	M41	100	88	M71	100	36
M12	100	76	M42	98	38	M72	99.3	38.7
M13	100	56.7	M43	100	60.7	M73	99.3	29.3
M14	100	59.3	M44	96	62	M74	100	62
M15	100	56	M45	100	54	M75	100	45.3
M16	99.3	37.3	M46	99.3	63.3	M76	100	66
M17	98.7	34.7	M47	99.3	70	M77	100	68.7
M18	100	31.3	M48	100	38	M78	99.3	40
M19	100	47.3	M49	99.3	66	M79	100	55.3
M20	99.3	60.7	M50	98.7	52	M80	100	26
M21	100	70	M51	100	46.7	M81	99.3	50
M22	98.7	48.7	M52	100	26.7	M82	100	22
M23	99.3	70.7	M53	99.3	58	M83	98.7	13.3
M24	100	36	M54	100	48	M84	100	53.3
M25	98.7	33.3	M55	100	64	M85	100	9.3
M26	98	21.3	M56	100	70.7	M86	100	47.3
M27	100	32	M57	98.7	57	M87	100	40.7
M28	99.3	61.3	M58	100	24	M88	98	80.7
M29	98.7	25.3	M59	100	64	M89	100	80
M30	98.7	19.3	M60	99.3	84	TH899	100.	38

Notes: G stands for rice genotype; GP stands for germination percentage. Values are the means of three replications.

**Table 3 biology-11-00126-t003:** K^+^ and Na^+^ content and K^+^/Na^+^ ratio of mutant lines and TH899 under control and saline–alkaline stress conditions.

Genotype	K^+^ Content (mM/g)		Na^+^ Content (mM/g)		K^+^/Na^+^	
	Control	Stress	Control	Stress	Control	Stress
M5	55.5 ± 1.1 ab	16.5 ± 0.7 ac	10.6 ± 0.1 b	25.5 ± 1.0 ab	5.2 ± 0.0 b	0.65 ± 0.1 ab
M41	55.1 ± 4.2 ab	21.2 ± 0.8 a	11.2 ± 0.5 b	27.9 ± 0.8 ab	4.9 ± 0.3 b	0.8 ± 0.0 a
M60	45.7 ± 4.1 b	15.1 ± 1.2 bc	11.7 ± 0.5 b	26.9 ± 1.8 ab	4.0 ± 0.5 b	0.6 ± 0.1 ab
M88	65.7 ± 3.1 a	16.4 ± 0.5 ac	11.9 ± 0.8 ab	27.5 ± 0.9 ab	5.6 ± 0.5 ab	0.6 ± 0.0 ab
M89	52.6 ± 0.7 ab	18.1 ± 0.9 ab	11.1 ± 0.6 b	24.2 ± 0.9 b	4.8 ± 0.2 b	0.8 ± 0.1 a
TH899	54.1 ± 1.0 ab	13.0 ± 1.0 c	14.9 ± 0.5 a	32.7 ± 1.6 a	3.7 ± 0.2 a	0.4 ± 0.0 b

Notes: Values are the means and standard errors of four replications. M5, M41, M60, M88, and M89 are highly tolerant mutant lines, and TH899 is the control. Different letters indicate significant differences in K^+^ and Na^+^ and K^+^/Na^+^ ratio between different rice genotypes at the same stress (*p* < 0.05, Tukey’s test).

**Table 4 biology-11-00126-t004:** Yield components of mutant lines under control and saline–alkaline stress conditions in 2019.

G	PH (cm)		PL (cm)		PN		SP		1000-GW (g)		PSF (%)	
	Control	Stress	Control	Stress	Control	Stress	Control	Stress	Control	Stress	Control	Stress
M5	101.3 ± 3.9 a	98.3 ± 4.0 a	19.5 ± 0.6 ab	20.2 ± 0.2 a	14 ± 1.0 a	10 ± 0.7 a	152 ± 5.7 a	130 ± 0.8 ab	25.05 ± 0.4 ab	23.60 ± 0.1 bc	94.5 ± 0.3 ab	93.8 ± 0.9 a
M41	108.3 ± 3.3 a	103.7 ± 1.1 a	16.8 ± 1.0 b	19.7 ± 0.4 a	16 ± 2.5 a	12 ± 0.4 a	98 ± 6.6 b	109 ± 7.7 b	26.18 ± 0.1 a	24.95 ± 0.1 ab	95.7 ± 0.5 ab	92.9 ± 2.0 a
M60	115.7 ± 4.0 a	102.7 ± 1.9 a	21.4 ± 0.2 a	21.3 ± 0.1 a	15 ± 1.5 a	11 ± 0.6 a	158 ± 9.6 a	152 ± 7.7 a	23.35 ± 0.6 b	20.83 ± 0.4 d	90.6 ± 2.1 b	91.9 ± 0.3 a
M88	106.0 ± 2.6 a	100.0 ± 1.7 a	21.1 ± 0.7 a	20.0 ± 0.2 a	14 ± 0.5 a	13 ± 1.0 a	164 ± 12.8 a	150 ± 2.6 a	23.30 ± 0.2 b	23.73 ± 0.3 abc	96.9 ± 0.6 a	97.7 ± 0.5 a
M89	107.0 ± 2.1 a	104.3 ± 2.6 a	20.2 ± 0.5 ab	19.4 ± 0.9 a	12 ± 0.9 a	12 ± 0.7 a	150 ± 10.5 a	120 ± 10.3 ab	26.12 ± 0.2 a	25.32 ± 0.3 a	96.8 ± 0.3 a	97.2 ± 0.3 a
TH899	105.7 ± 2.6 a	97.7 ± 1.8 a	19.8 ± 0.1 ab	18.5 ± 1.0 a	13 ± 0.9 a	11 ± 0.4 a	153 ± 0.7 a	137 ± 8.1 ab	23.63 ± 0.5 b	22.85 ± 0.4 c	95.4 ± 1.2 ab	94.7 ± 1.9 a

Notes: G indicates rice genotype; PH indicates plant height; PL indicates panicle length; PN indicates panicle number per plant; SP indicates spikelet per panicle; 1000-GW indicates 1000-grain weight; and PFS indicates percentage of filled spikelet. Values are the means and standard errors of four replications. M5, M41, M60, M88, and M89 are highly tolerant mutant lines, and TH899 is the control. Different letters indicate significant differences in yield components between different rice genotypes at the same stress (*p* < 0.05, Tukey’s test). The “a” and “ab” contain the same letter "a", indicating that there is no significant difference; the "b" in “ab” indicates that there is no significant difference and other groups containing the letter b (such as “bc”), but there is a significant difference between “a” and “bc”.

**Table 5 biology-11-00126-t005:** Yield components of mutant lines under control and saline–alkaline stress conditions in 2020.

G	PH (cm)		PL (cm)		PN		SP		1000-GW (g)		PSF (%)	
	Control	Stress	Control	Stress	Control	Stress	Control	Stress	Control	Stress	Control	Stress
M5	96.1 ± 1.6 c	84.8 ± 0.6 c	20.8 ± 0.2 a	18.3 ± 0.4 ab	10 ± 0.4 a	8 ± 0.6 a	149 ± 9.5 a	112 ± 3.2 ab	25.89 ± 0.5 a	23.99 ± 0.1 ab	97.1 ± 1.0 a	96.2 ± 1.4 a
M41	101.6 ± 1.3 ac	90.5 ± 1.2 bc	19.7 ± 0.4 a	17.7 ± 0.4 b	9 ± 0.7 a	7 ± 0.4 a	119 ± 4.8 a	103 ± 5.8 b	27.04 ± 0.6 a	25.30 ± 0.4 a	93.4 ± 0.9 a	91.45 ± 3.7 a
M60	107.1 ± 2.3 ab	95.0 ± 1.3 ab	21.4 ± 0.3 a	19.8 ± 0.1 a	11 ± 0.4 a	7 ± 0.7 a	172 ± 9.1 a	133 ± 2.5 a	23.11 ± 0.3 b	22.02 ± 0.6 bc	92.9 ± 1.4 a	95.4 ± 0.8 a
M88	94.2 ± 1.8 c	85.3 ± 2.7 c	20.3 ± 0.8 a	18.4 ± 0.4 ab	9 ± 0.4 a	7 ± 0.5 a	162 ± 15.1 a	132 ± 8.5 a	22.21 ± 0.4 b	22.18 ± 0.3 bc	92.4 ± 1.1 a	96.4 ± 0.7 a
M89	98.3 ± 3.1 bc	89.3 ± 0.3 bc	19.2 ± 0.5 a	17.5 ± 0.3 b	10 ± 0.4 a	8 ± 0.9 a	136 ± 15.4 a	108 ± 4.2 ab	25.49 ± 0.1 a	23.81 ± 0.9 ab	96.3 ± 0.9 a	93.2 ± 1.5 a
TH899	110.9 ± 0.9 a	98.5 ± 3.3 a	16.7 ± 0.2 b	15.3 ± 0.3 c	9 ± 0.9 a	6 ± 0.6 a	166 ± 9.4 a	114 ± 3.9 ab	22.32 ± 0.2 b	20.40 ± 0.1 c	95.9 ± 0.9 a	97.3 ± 0.3 a

Notes: G indicates rice genotype; PH indicates plant height; PL indicates panicle length; PN indicates panicle number per plant; SP indicates spikelet per panicle; 1000-GW indicates 1000-grain weight; and PFS indicates percentage of filled spikelet. Values are the means and standard errors of four replications. M5, M41, M60, M88, and M89 are highly tolerant mutant lines, and TH899 is the control. Different letters indicate significant differences in yield components between different rice genotypes at the same stress (*p* < 0.05, Tukey’s test).

## Data Availability

The data presented in this study are available in the article.

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
