# Peer review of "Evaluation of the Saline–Alkaline Tolerance of Rice (Oryza sativa L.) Mutants Induced by Heavy-Ion Beam Mutagenesis"

_biology, 2022, doi:10.3390/biology11010126_

Round 1

Reviewer 1 Report

The results are interesting. However, how this method does affect the nutritional and flavor properties? I think, this is also interesting and very important.

What other traits important to the mode of cultivation are affected by this method? Is the pollen of the flowers altered? Will the mutation affect other rice varieties grown adjacent to it?

Conclusions: The answer to the questions posed in the research hypothesis was not provided. Please complete. Please also provide specific research applications.

Reviewer 2 Report

The authors present a cogent study on saline and alkaline tolerance of rice (Oryza sativa L.) mutants induced by heavy-ion beam mutagenesis. The work is carried and presented in a professional way with attention to detail. There are a few small concerns before this paper is ready to publish.

I would expect supporting information with sequences etc to be provided.

The abstract needs to be shortened, it needs to be presented as a succinct summary, so the reader knows if they want to read the paper or not. The abstract is abit repetitive in the current form and needs to be shortened by 40-50%, with key points highlighted.

Small formatting issues eg line 138/139

Table 2 – ‘100.0’ can be written as ‘100’ and in the text eg line 184 '99.51' should be written as '99.5' - these levels of accuracy are not needed or proved at the current time.

The foot note on table 3 needs further explanation - the 'ab' etc after their letters is abit confusing. This is the same for table 4 and 5, these tables need to be cleaned up and formatted correctly. It is good data presented in a messy way and it detracts from the quality work.

A through English check would be useful.

Reviewer 3 Report

Thank you for this well-written manuscript.  -  Throughout your manuscript, you are pointing out that heavy ion beam radiation is a convenient method to create stress tolerant mutants. Well, you know that it is not the only method. Therefore, I suggest adding a bit more information on alternatives. This would help stakeholders taking a decision.
